# Post-Fire Soil Nutrient Dynamics in *Seriphium plumosum* L. Encroached Semi-Arid Grassland of Gauteng Province, South Africa

Hosia T. Pule [1,2,*,†], Julius T. Tjelele [1] and Michelle J. Tedder [2]

1    Agricultural Research Council, Animal Production, Range and Forage Science, Irene 0062, South Africa; jtjelele@arc.agric.za

2    School of Life Sciences, College of Agriculture, Engineering and Science, University of Kwa-Zulu Natal, Scottsville 4041, South Africa; tedder@ukzn.ac.za

*    Correspondence: gpule@arc.agric.za; Tel.: +27(0)-12-672-9343

†    This paper is a part of the PhD Thesis of Hosia T. Pule, presented at the University of KwaZulu-Natal (UKZN), South Africa.

**Abstract:** *Seriphium plumosum* L. is an indigenous unpalatable shrub that occurs in fire-prone semi-arid South African grassland areas, yet research proposes the use of fire to control its encroachment of rangelands. This study investigated the interaction effects of burning and soil depth on components of soil fertility. Soil samples were collected from the surface (<10 cm) and subsurface (>10 ≤ 20 cm) soil, before and after burning in randomly selected paired subplots (25 m × 25 m), with six replicates. Data was analysed as a randomised complete block design, with repeated measures (before and after burning) in a 2 × 2 factorial analysis of variance (ANOVA) using generalised linear model (GLM) procedures. Components of soil fertility measured (K, Ca, Mg, Org C, P, pH and TN) showed a significant decrease with increasing soil depth both before and after burning, except for K and P, which were significantly higher in surface soils after burning. The results showed that the response of soil nutrients to fire depends on the temperature tolerance threshold of individual soil nutrient elements. Increasing surface soil available K and P concentrations after burning may improve the conditions for *S. plumosum* encroachment, with implications for similar environments and species worldwide.

**Keywords:** burning; combustion; desertification; encroachment; grassland

## 1. Introduction

Global drylands have evolved with fires [1], which occur from both natural and anthropogenic ignition [2]. Africa is referred to as the "fire continent" [3], with fires occurring from January to April in West Africa and from July to October in eastern and southern Africa [4]. These fires affect the soil, flora and fauna [5], thus shaping the global biome's distribution and maintaining its structure and function [6]. Although fire is a cost-effective management tool to control woody plant encroachment [7], and increase forage production [8], its effectiveness in semi-arid grasslands is questioned [9–11], especially without browsers [12]. Hence, understanding the role of fire in shaping both vegetation and soil structure and functioning is key to managing woody plant encroachment, especially in fire-prone semi-arid grasslands.

Research on post-fire soil nutrient concentrations in semi-arid rangelands is lacking [13], yet it is important in biogeochemical cycles and the ecology of microbial, plant, and animal communities [14], among others. Soil nutrients also play an important role in maintaining the structure and function of grassland ecosystems [15], but the interaction effect of fire and soil fertility in driving rangelands' ecosystem structure and function remains underresearched. Consequently, this limits our understanding of the role of fire in soil nutrient cycling and on woody plant encroachment control, especially of species such as *S. plumosum*, which thrive in fire-prone areas.

The effects of fire on soils may be brief, extended, or permanent, depending on the soil property concerned, the frequency and severity of the fire [16], the type of burned vegetation, and the regional climatic conditions [17]. These effects on soil nutrients are usually restricted to the first few centimeters of the topsoil [18,19] and are driven by fire temperature. Consequently, understanding how soil nutrients respond to fire may provide essential knowledge about biogeochemical cycles, ecosystem succession, and general grassland management [20].

*Seriphium plumosum* L., also known as slangbos or bankrupt bush, is a multi-stemmed encroacher shrub in the Asteraceae family, indigenous to South Africa [21,22]. Its encroachment reduces grass production [23–25] and biodiversity [26]. While chemical and mechanical control measures for *S. plumosum* encroachment are expensive and labour intensive [24], the use of fire in controlling *S. plumosum* encroachment remains a subject of much debate [27]. In addition, research on post-fire soil nutrient dynamics of *S. plumosum* encroached areas is lacking [28], yet it is critical to aid land managers in predicting ecosystem recovery responses post fire [29], and to quantify land degradation processes and post-fire restoration plans [30].

This study explored the short-term interaction effect of burning (before and after fire) and soil depth (<10 cm and 10–20 cm) on components of soil fertility (potassium (K), phosphorus (P), magnesium (Mg), total nitrogen (TN), sodium (Na), calcium (Ca), soil organic carbon (SOC) and pH) in *S. plumosum* encroached semi-arid grassland communities in Gauteng Province. It is hypothesized that (1) intense fires on surface soils will lead to the combustion of SOC and TN loss caused by volatilization, and (2) accumulating ash post-fire will increase the general surface soil nutrients (K, P, Mg, Na, Ca and pH) content.

## 2. Materials and Methods

### 2.1. Study Area

The experiment was conducted on Carletonville Dolomite Grassland (CDG) [31], situated in Gauteng Province, South Africa. The experimental sites are located on a gentle (approx. 10°) north-facing slope [32], at an elevation of approximately 1614 m above sea level. Rain falls almost exclusively in the summer (October–April), with a mean of 593 mm per annum [31]. The average minimum and maximum summer and winter temperatures for the CDG are 15.4–30 °C and 6–21 °C [31], respectively.

*Seriphium plumosum* L. density and canopy size at the experimental site were 1 plant/2.17 m$^2$ ($\pm$2.39 (SEM)) and 1.39 m$^2$ ($\pm$0.11), respectively. The soils are predominantly from Dolomite and Chert of the Malmani subgroup, which support mostly shallow Mispah and Glenrosa soil forms typical of the Fa land type [31]. These soils had a mean silt and clay content of 8.66% ($\pm$0.96) and 28% ($\pm$0.29), respectively [32]. The land use is mainly cattle grazing and the area burns approximately once every three years.

### 2.2. Sampling Design

The study used a factorial design consisting of burning (before and after fire) and soil depth (<10 cm and 10–20 cm), with six replicates [32]. Soil samples were randomly collected from burned subplots of 25 m × 25 m each before and after burning. The sampling was carried out on the same subplots before and after burning to allow for comparison. There was a minimum distance of 10 m between the randomly selected subplots. These experimental plots were in an enclosed camp, free from human disturbance and cattle grazing. Furthermore, soil sampling before and after fire was performed before rainfall to avoid the effect of rainfall on components of soil fertility.

### 2.3. Soil Sampling

Since southern African fires occur mostly between July and October [6], soil samples (n = 20) were randomly collected an hour before burning (August) and a month after burning (September), with each fire lasting for the duration of approximately 30–45 min. These soil samples were collected from the surface (<10 cm) (n = 10) and subsurface (10–20 cm) soil layers (n = 10), and pooled together per soil depth. The total number of soil samples before

and after burning, was 36 and 36, respectively. The soils were analysed for K, P, Mg, TN, Na, Ca, SOC and pH. Soil P was determined using a P Bray No. 1, following Mallarino and Blackmer [33]. Soil K, Ca, Mg and Na content were determined using the soil ammonium acetate extraction method [34]. Soil organic carbon was determined following the Walkley Black [35] procedure, while TN was determined using a Total Nitrogen digester. Soil pH was determined using water, with a 2:5 soil:water ratio [36].

*2.4. Statistical Analysis*

The effects of burning (before and after fire), soil depth (<10 cm and 10–20 cm), and their interactions on components of soil fertility were analysed as a randomised complete block design, with repeated measures (before and after fire) in a 2 × 2 factorial analysis of variance (ANOVA) using generalized linear model (GLM) procedures. Burning and soil depth were analysed as independent variables and components of soil fertility were analysed as dependent variables. Soil Mg, TN, Ca and pH values were log-transformed to meet the normality and homogeneity assumptions of ANOVA, but this transformation was not necessary for K, P, Na and SOC. The data was analysed using SPSS, version 15 of 2016. When the ANOVA produced significant results, the means were compared using Tukey's HSD test, and the differences were declared significant at $p < 0.05$.

## 3. Results

Burning significantly affected soil K, Na, and P ($p < 0.05$). Similarly, soil depth had a significant effect on soil K, Ca, Mg, Org C, P, pH and TN ($p < 0.05$), and the interaction between burning and soil depth significantly affected soil K, Ca, Mg, Org C, P and TN ($p < 0.05$; Table 1).

**Table 1.** ANOVA results for the main factor of burning, soil depth, and their interaction on components of soil fertility.

|  |  | K (mg/kg) |  | Na (mg/kg) |  | Ca (mg/kg) |  | Mg (mg/kg) |  | SOC (%) |  | P (mg/kg) |  | pH |  | Total N (%) |  |
|---|---|---|---|---|---|---|---|---|---|---|---|---|---|---|---|---|---|
|  | DF | F | P | F | P | F | P | F | P | F | P | F | P | F | P | F | P |
| Burning | 1 | 6.92 | **0.01** | 12.80 | **0.01** | 0.51 | 0.48 | 3.57 | 0.06 | 1.20 | 0.28 | 15.30 | **0.01** | 2.46 | 0.12 | 0.24 | 0.63 |
| Soil depth | 1 | 55.49 | **0.01** | 1.01 | 0.32 | 30.36 | **0.01** | 18.39 | **0.01** | 44.10 | **0.01** | 121.35 | **0.01** | 36.22 | **0.01** | 88.84 | **0.01** |
| Time × Soil depth | 1 | 4.44 | **0.04** | 0.04 | 0.85 | 10.10 | **0.01** | 14.67 | **0.01** | 3.83 | **0.05** | 6.63 | **0.01** | 3.69 | 0.59 | 4.23 | **0.04** |

Significant values are shown in bold: K; P; Mg; TN Na; Ca; SOC.

Burning significantly increased soil K (before: 94.74 ± 3.36 (SEM), after: 116.20 mg/kg ± 6.74), Na (before: 4.33 mg/kg ± 0.21, after: 6.27 mg/kg ± 0.41) and P (before: 3.23 mg/kg ± 0.15, after: 4.34 mg/kg ± 0.27). Soil K (surface: 127.64 mg/kg ± 4.77, subsurface: 76.15 mg/kg ± 2.65), Ca (surface: 154.88 mg/kg ± 7.02, subsurface: 90.37 mg/kg ± 4.28), Mg (surface: 60. 70 mg/kg ± 3.08, subsurface: 39.72 mg/kg ± 1.78) and Org C (surface: 1.54 mg/kg ± 0.02, subsurface: 1.28 mg/kg ± 0.01) were significantly higher in the surface compared to subsurface soils. Similarly, soil P (surface: 4.99 mg/kg ± 0.01, subsurface: 2.29 mg/kg ± 0.13), pH (surface: 4.91 ± 0.022, subsurface: 4.71 ± 0.01) and TN (surface: 0.06% ± 0.0009, subsurface: 0.05% ± 0.067) content was significantly higher in surface soils compared to subsurface soils. There was a significant interaction effect of burning × soil depth on soil K, Ca, Mg, Org C, P and TN ($p < 0.05$). However, these effects were pronounced on soil available K and P concentrations, which were significantly higher after fire than before fire on the surface soil ($p < 0.05$; Figure 1).

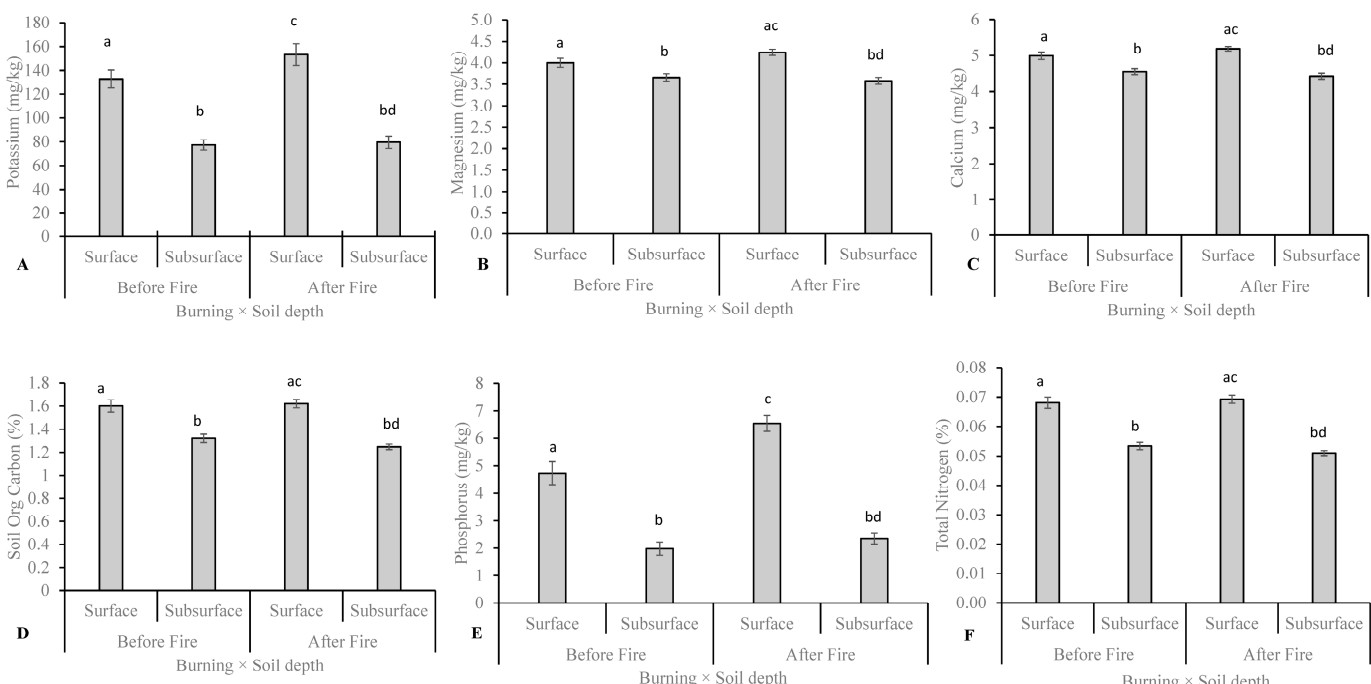

**Figure 1.** Interaction effect of burning × soil depth on the concentration of soil P (**A**), Ca (**B**), Mg (**C**), SOC (**D**), P (**E**) and TN (**F**) concentrations. The same letters on the bars mean that $p > 0.05$.

## 4. Discussion

The fire increased soil available K, Na, and P, but not Ca, Mg, SOC and TN concentrations and pH levels. Soil fertility (K, P, Mg, TN, Ca, SOC and pH) decreased with soil depth, while fire × soil depth increased soil K and P concentrations post fire, especially in the surface soils.

Fire has the potential to mobilize nutrients, so depending on its intensity and frequency, it may increase or decrease nutrient availability in the ecosystem [37]. Nitrogen and SOC have a temperature tolerance threshold as low as 200 °C [38], with half of nitrogen volatilization occurring at a temperature of 500 °C. This suggests that the prescribed fire used in this study was relatively hot (>200 °C) to cause combustion of organic matter and nitrogen loss by volatilization [39]. The fire increased soil available P, K and Mg concentrations [40–42]. Soil available K and P have moderate fire sensitivity and a fire tolerance threshold of 774 °C [43], hence their increase with fire in this study. The higher heat tolerance thresholds of soil available Mg, Ca, and Mn of 1107 °C, 1484 °C, and 1962 °C [44], respectively, in part explaining their stability post-fire in this study. Gowlett et al. [45] also found temperatures for the grass-brush communities in South African semi-arid areas to peak at 500 °C, which may suggest that the prescribed fire used in this experiment was probably above 200 °C but lower than 774 °C. Hence, the observed changes occurred exclusively on components of soil fertility with a threshold of 774 °C in this study.

Generally, the responses of individual components of soil fertility to fire are inconsistent and non-universal [46], in part because each has an inherent temperature tolerance threshold [44], among other factors. Soil available K and P increase with fire [47], especially in the surface soils, as shown in this study. Ando et al. [48] reported increasing levels of available nutrients such as P and exchangeable K following fire. Although soil-available P responds differently to fire, substantial amounts of readily available P are found in the ash and on the soil surface immediately following a fire [44]. Soil available P [49] and K [50] are the most important soil parameters influencing woody plant cover after mean annual precipitation and fire. Consequently, their increase after fire in surface soils may contribute to facilitating *S. plumosum* encroachment in the semi-arid grassland communities of Gauteng Province, South Africa.

A combination of combustion and heat transfer that produces a sharp temperature gradient in the surface soil profile [51] might contribute to explaining the association between *S. plumosum* and high-temperature and/or fire-prone areas [52]. Fire may contribute to controlling woody plant encroachment [53,54] while improving soil nutrient status, especially soil K and P concentrations, which favour the conditions for *S. plumosum* germination from a soil seedbank [21], which explains its encroachment on rangelands. The reduction in SOC and TN concentrations post-burning [39] and the lack of their change with fire in this study favours the condition for *S. plumosum* encroachment [24]. These results suggest that fire is an important factor driving *S. plumosum* distribution and occurrence, hence its association with fire-prone areas. There is, however, a need for more long-term research on the effects of fire intensity and frequency on components of soil fertility and *S. plumosum* population dynamics in South African semi-arid grasslands.

## 5. Conclusions

This study has shown that fire did not affect SOC and TN, but improved soil available K and P concentrations, especially in the surface soil. These conditions may favour *S. plumosum* germination from soil seed banks, seedling recruitment and, consequently, its potential encroachment on rangeland communities. Furthermore, the ability of *S. plumosum* to coppice or resprout after fire might improve with improved conditions for growth. An understanding of the role of fire intensity and severity on soil fertility, soil seed banks, and the control of *S. plumosum* encroachment is also lacking. Hence, there is a need for more long-term research on how multiple interacting factors in space and time contribute directly or indirectly to causing *S. plumosum* encroachment in South African grassland communities.

**Author Contributions:** Study Conceptualization, H.T.P., J.T.T. and M.J.T. Methodology, H.T.P., J.T.T. and M.J.T. Validation, J.T.T. and M.J.T. Formal Analysis, H.T.P. Writing—Original Draft Preparation, H.T.P. Writing—Review and Editing, H.T.P., J.T.T. and M.J.T. Supervision, J.T.T. and M.J.T. Project Administration, H.T.P., J.T.T. and M.J.T. Funding Acquisition, J.T.T. All authors have read and agreed to the published version of the manuscript.

**Funding:** The work was funded by the National Research Foundation (Grant No: 116280), South Africa. Julius Tjelele received this grant or research support from the National Research Foundation (NRF), South Africa.

**Institutional Review Board Statement:** Ethical review and approval were waived for this study because it is not applicable for studies not involving humans or animals.

**Data Availability Statement:** The datasets generated during and/or analysed during the current study are available from the corresponding author and can be made available on reasonable request.

**Acknowledgments:** We are very grateful to the Agricultural Research Council: Animal Production Range and Forages Sciences team for their assistance during data collection. We are thankful to the ARC: ISCW (Noluthando Sotaka) for helping us with soil analysis. We would also like to acknowledge and thank the ARC: AP-Biometry (Eric Mathebula), for assistance with data analysis. This work was funded by the National Research Foundation (Grant No: 116280), South Africa.

**Conflicts of Interest:** The authors declare no conflict of interest. The funders had no role in the design of the study; in the collection, analyses, or interpretation of data; in the writing of the manuscript; or in the decision to publish the results.

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
