# Peer review of "Post-Fire Soil Nutrient Dynamics in Seriphium plumosum L. Encroached Semi-Arid Grassland of Gauteng Province, South Africa"

_agriculture, doi:10.3390/agriculture13101971_

Round 1

Reviewer 1 Report (Previous Reviewer 2)

My opinion is that the research has some interest and I would like to help authors to improve the paper. As it is now, it has some deficiencies and can be improved.

The first comment is about the title as it is written: “Post-fire soil nutrient dynamics in Seriphium plumosum L. en-2 croached semi-arid grassland of Gauteng Province, South Af- rica”, I believe that if it is possible, not to cut Af-rica, better A-frica or Afri-ca. But his is only a suggestion.

Regarding the content of the article, I would like to ask some comments and minor suggestions.

In the introduction fire is presented as a control tool. For instance, in line 54: the use of fire in controlling S. plumosum… I do not know if there is or not an alternative to the farmers. It seems that there is not and in such a case, fire is the only option and the effects on soil properties should occur.

In materials and methods, please indicated the meaning of SEM. Regarding the topography, is the area a plateau or a piedmont or terraces? And moreover, in this section I suggest to add the type of soil according to the World Reference Base for soil resources (WRB) or Soil Taxonomy (USDA).

The results, are more or less expected and they respond to the hypothesis of the work. I agree with the results obtained. However, as a minor detail, the format of the table 1 would be improved.

In the discussion, some comments should be explained and try to indicate why these results happened. For instance, at the beginning “Fire increased soil K, Na, and P, but not Ca, Mg, SOC, pH and TN concentrations”. I prefer not to talk about concentration when talking about pH.

Moreover, in the discussion, it is not easy to justify why Ca and Mg diminished in the soil. I recommend that available Ca and Mg diminished in the soil, not Ca and Mg. The difference is very important because Ca and Mg could be in form or oxides CaO, for instance, and total Ca would be increased but not the availability of Ca. As a general comment that should be considered, talk about availability of K, P, etc., because as it is written in the text, someone can think that you are talking about total concentrations in soil. Probably, only organic carbon and nitrogen would be reduced, depending on the fire intensity.

The conclusions would be centred in available nutrients. I agree that more research is needed.

References are good, of course they can be enriched, but I believe they are enough. Please check the format and the style guide of the journal.

In general, I missed the comments and temperatures reached by the fire, to estimate the intensity. But, if you do not have them, please try to comment and indicate how the intensity was.

Some minor details about the format of the text, the spaces between paragraphs and others, would be checked later if this article is accepted.

Author Response

Dear Reviewer,

Thank you for your valuable comments, suggestions and inputs aimed at improving the overall quality of our manuscript. All your comments and suggestions are addressed to the best of my ability.

Kind Regards

Reviewer 2 Report (New Reviewer)

On the one hand, I appreciate the fact that you are short and concise regarding the introduction, but I do not understand the following aspects:

1. What is the difference in germination of the S. plumosum species in these burned plots and what is outside the plots, where there is no burned land? That is, in what proportion did the species appear after the land was burned compared to the land where it was not?

2. What are the elements that maintain the S. plumosum species?

3. After only one year, you certainly cannot draw conclusions, as you mentioned, but you should also see the growth capacity of the species in the two land conditions (burned/unburned).

Author Response

Dear Reviewer,

Thank you for reviewing my manuscript. I found your review very helpful and helped in improving the overall quality of our manuscript.

Kind Regards

Reviewer 3 Report (New Reviewer)

(1) In the Introduction section, it is suggested to add a description of the innovation of this paper.

(2) The authors only selected soil at 0-20cm depth for sample collection and analysis, and did not consider the effect of fire on soil greater than 20cm depth, please explain the reason.

(3) The authors' sampling time was August-September, please explain whether the sampling time was affected by rainfall in the Methods section. In addition, whether fire experimental design at different times at the end of the rainy season is likely to affect the results of the study (i.e. fire field experiments just after or just before the rainy season). Please explain.

(4) The study area is sloping terrain, whether having sloping terrain will affect the results of the study, please explain in the discussion.

Minor editing of English language required

Author Response

Dear Reviewer,

Thanks for accepting to review my manuscript. I found your review very insightful and helped us to improve the overall quality of this manuscript. I have addressed all your concerns and comments to the best of my ability. I have also send the paper for English proof reading to address your concerns regarding the us of English language...

Looking forward to hearing from you.

Regards 

Round 2

Reviewer 1 Report (Previous Reviewer 2)

Thanks for improving the article. I have some comments but I do not know if you are going to be able to solve.

The type of soil using an international classification (WRB or Soil Taxonomy)

The temperature reached in the soil during burning. 

The format of the sub-sections and the references, for instance years in bold letter.

Probably, you do not have the solutions to these comments, but please, consider them as compulsory for future articles where the soil is the key.

Author Response

Dear Reviewer,

Thank you so much for reviewing my paper and for your great comments and suggestions. Indeed, they are valid and have helped me improve the overall quality of our manuscript.

I have attached the revision letter below to further assist you in reviewing this revised version.

Looking forward to hear from you.

Kind Regards

Dr Pule

This manuscript is a resubmission of an earlier submission. The following is a list of the peer review reports and author responses from that submission.

Round 1

Reviewer 1 Report

The details are as follows.

Reviewer 2 Report

In general, my point of view about the article is that it is well designed, good experiment done and interesting results, close to that expected considering the use of fire for the improvement of cultivated areas.

I have some comments about the article. The first one is about the use of fire to control this specie. Is it necessary or do you have alternatives?

I understand that some nutrients will be favoured by fire but I do not understand the case of Ca and Mg. I do not know why there is no increment from plant burned tissues. This need a better explanation.

Obviously, it was expected a reduction of organic matter and nitrogen. Although the intensity of fire is a key factor.

The experiment seems in part, not finished. As authors indicted at the end of the discussion: “a need for more long-term research on the effects of fire intensity and frequency on components of soil fertility and S. plumosum population dynamics on South African semi-arid grasslands”.

To have better conclusions, you will need to measure the intensity of fire, unless the temperature achieved. Without this data, it is so difficult to evaluate the results obtained. In fact, which seems important is that the plant tested has the ability to increase its expansion on the soil after fire, but this also needs to be demonstrated counting the plants growth in the following season after the fire. Probably authors can check this in the experimental plots.

Minor comments:

Try to cut the words in a way that they can be easily read. For instance, in the title of the article “Af-rica”, maybe Afri-ca or A-frica. However, I understand that this is a matter of the final format of the text. Other examples (lines 88, 97, 193,…).

Check the format of the references, for instance in line 36: [11;12; 13].

The way as the table is presented can be improved and probably, it is not necessary the use of bold letter in the table 1, for instance in the numbers given.